# *Chaenomeles sinensis* Extract Ameliorates Ovalbumin-Induced Allergic Rhinitis by Inhibiting the IL-33/ST2 Axis and Regulating Epithelial Cell Dysfunction

**DOI:** 10.3390/foods13040611

**Published:** 2024-02-18

**Authors:** Juan Jin, Yan Jing Fan, Thi Van Nguyen, Zhen Nan Yu, Chang Ho Song, So-Young Lee, Hee Soon Shin, Ok Hee Chai

**Affiliations:** 1Department of Anatomy, Jeonbuk National University Medical School, Jeonju 54896, Republic of Korea; jinjuan0619@gmail.com (J.J.);; 2Institute for Medical Sciences, Jeonbuk National University Medical School, Jeonju 54896, Republic of Korea; 3Department of Food Biotechnology, University of Science and Technology (UST), Daejeon 34113, Republic of Korea; 4Division of Food Functionality Research, Korea Food Research Institute (KFRI), Wanju 55365, Republic of Korea

**Keywords:** allergic rhinitis, airway inflammation, *Chaenomeles sinensis*, cytokines, immunoglobulins

## Abstract

*Chaenomeles sinensis* has traditionally been used as an herbal medicine due to its characteristics that protect against inflammation, hypertension, and mutagenesis. However, the effect of *Chaenomeles sinensis extract* (CSE) on allergic rhinitis (AR) and its underlying mechanisms have yet to be thoroughly investigated. The current study explored the likely effect of CSE on AR in an ovalbumin (OVA)-induced AR mouse model. To this end, OVA-specific immunoglobulins, nasal symptoms, cytokine production, the infiltration of inflammatory cells, and nasal histopathology were assessed to determine the role of CSE against AR. The supplementation of CSE was found to suppress OVA-specific IgE, while OVA-specific IgG2a was increased in the serum. Further, CSE ameliorated the production of T helper type 2 (Th2) cytokines whereas it increased Th1 cytokine levels in nasal lavage fluid. Moreover, the CSE treatment group exhibited significant inhibition of IL-33/ST2 signaling. Subsequently, CES reversed the OVA-induced enhancement of epithelial permeability and upregulated E-cadherin, thus indicating that CES plays a protective role on epithelial barrier integrity. Altogether, the oral administration of CSE effectively controlled allergic response by restricting the buildup of inflammatory cells, enhancing nasal and lung histopathological traits, and regulating cytokines associated with inflammation. Collectively, the results show that the supplementation of CSE at different doses effectively regulated AR, thus suggesting the therapeutic efficiency of CSE in suppressing airway diseases.

## 1. Introduction

Allergic rhinitis (AR) is a prevalent allergic ailment that primarily affects the nasal cavity. It is distinguished by inflammation of the nasal passages. The condition is typically referred to as hay fever, and it is characterized by an increased immunological response to various allergens, including pollen, dust mites, and mold spores. AR is frequently related to asthma, rhinosinusitis, allergic conjunctivitis, and adenoid hypertrophy [1]. AR is known to cause nasal irritation, congestion, and sneezing, meaning it plays a substantial role in the progression of asthma [2]. Therefore, AR management is key to reducing the risk of asthma development [3]. Typically, AR persists throughout one’s life, and it impacts approximately 10% to 30% of the global population [4]. AR can cause non-life-threatening complications such as emotional instability, sleeplessness, and reduced quality of life [5]. AR is activated when allergen-specific immunoglobulin E (IgE) and T helper 2 (Th2) cells detect the presence of allergens that have been absorbed from the environment. Numerous inflammatory cells (such as eosinophils and mast cells), cytokines, and additional regulatory molecules are also involved in the AR inflammatory process [6]. Increased allergen-specific IgE levels, along with a Th1/Th2 imbalance, have been recognized as the primary immune deviations contributing to AR [7]. In general, AR can be managed using steroidal and non-steroidal anti-inflammatory agents. However, these anti-inflammatory agents may result in side effects such as burning, mild irritation, dry mouth, drowsiness, occasional nosebleeds, and nose dryness [8,9,10]. Hence, it is essential to develop functional foods without side effects to avoid side effects and enhance the quality of life. Functional foods are harmless, and their efficacy might be identical to conventional treatments. Moreover, individuals tend to rapidly adopt functional foods with anti-inflammatory properties, as they are safe and generally have fewer side effects. Consuming functional foods with high polyphenol content can reduce the infiltration of inflammatory cells caused by allergens, lower serum IgE levels, and limit the production of inflammatory cytokines and histamine from mast cells, thereby resulting in anti-inflammatory effects [11,12]. Moreover, previous reports suggest that polyphenols can impede the production of pro-inflammatory cytokines, such as interleukin-6 (IL-6), tumor necrosis factor-alpha, and interleukin-8 (IL-8). This helps reduce the inflammatory activities in the respiratory passages. Moreover, polyphenols can hinder the activation of nuclear factor-kappa B (NF-κB), which is a pivotal regulator of inflammation [13,14,15,16]. This action contributes to lowering the magnitude of the inflammatory responses. It is therefore crucial to identify polyphenol-rich functional foods to aid in the development of safe and effective treatments to prevent airway hyperresponsiveness and AR.

*Chaenomeles sinensis*, which is commonly known as Chinese quince or Chinese flowering quince, is a deciduous shrub that is native to China. This plant is predominantly cultivated for its aesthetic appeal due to its attractive blossoms, while its hard and astringent fruit is used in traditional Chinese medicine and as a culinary ingredient in East Asian cuisine. It is used either alone or with other medicinal plants to treat diarrhea, regurgitation, myalgia, and the common cold [17,18]. *C. sinensis* contains bioactive compounds such as triterpenes and phenolic compounds known to possess antimicrobial, anti-inflammatory, anti-hypertensive, neuroprotective, and anti-mutagenic effects [19,20,21,22,23,24]. In particular, terpenoids—such as thymol and menthol—display bronchodilatory characteristics [25,26], relax airways, and ease respiratory discomfort. They also regulate inflammatory pathways by suppressing the release of cytokines and the migration of leukocytes [27,28,29]. Moreover, phenols demonstrate antioxidant and anti-inflammatory properties by eliminating free radicals [30,31,32] and inhibiting pro-inflammatory enzymes such as cyclooxygenase [33,34]. Compounds like these reduce airway inflammation by reducing oxidative stress and cytokine production. These activities of phenols aid in easing respiratory issues by enhancing lung functioning and mitigating symptoms. It has also been reported that *C. sinensis* can inhibit oxidative damage [22], which is closely linked to the progression of chronic inflammation. Previous reports have suggested that *C. sinensis* is rich in polyphenol compounds [35], which potentially act against inflammation and oxidation. Interestingly, earlier studies have suggested that epigallocatechin gallate (EGCG), a significant catechin found in C. sinensis, boosts cellular antioxidant defenses by increasing the expression of antioxidant enzymes such as superoxide dismutase (SOD) and catalase [36]. These compounds synergistically act to protect cells and tissues from oxidative damage, thus contributing to the potential health benefits. These properties are of substantial importance in relation to the pathogenesis of respiratory disorders. However, the effect of *Chaenomeles sinensis* extract (CSE) on airway inflammation in AR and its underlying processes have yet to be thoroughly explored. Therefore, the current study aims to evaluate the influence of CSE on airway inflammation during AR in an OVA-induced AR mouse model.

## 2. Materials and Methods

### 2.1. Preparation of CSE

CSE was provided by the Korea Food Research Institute (Wanju-gun, Republic of Korea). Briefly, raw *C. sinensis* fruit was dried, pulverized, and extracted with 20× of 50% ethanol at ±50 °C for 6 h. Next, CSE was filtered via a cartridge filter. Following this filtration, CSE was concentrated at 50 °C under a vacuum using a Rotavapor R-210 (BÜCHI Labortechnik AG, Flawil, Switzerland). Then, CSE was freeze-dried and stored until use.

### 2.2. Analysis of CSE Using UPLC-Q-TOF-MS

The CSE was analyzed to assess the presence of bioactive compounds using a UPLC-Q-TOF-MS system (Waters, Milford, MA, USA). The bioactive compounds in CSE were isolated with the Acquity UPLC BEH C18 column (Waters, Milford, MA, USA); the column was equilibrated with water containing 0.1% formic acid and eluted with a gradient of acetonitrile containing 0.1% formic acid. All of the eluted samples were processed through the Q-TOF-MS system in positive electrospray ionization mode using the following instrument settings in the 50 to 1500 *m*/*z* range: cone voltage, 20 V; capillary voltage, 2.5 kV; nebulized gas, 900 L/h at a temperature of 100 °C in positive mode; and cone gas flow, 30 L/h. All analyses were performed using a locking spray to ensure accuracy and reproducibility. The compounds were identified referring to online databases. The identified compounds were quantitatively analyzed using the multiple reaction monitoring (MRM) mode of a UPLC-Q-TOF-MS system.

### 2.3. Animal Studies

Male five-week-old BALB/c mice were purchased from Damool Sciences, Daejeon, Korea. Mice (n = 6 mice/cage) were housed in fully climate-controlled, pathogen-free conditions with a 12 h:12 h L/D cycle. All animals had unrestricted access to food and water. Mice received standard care and maintenance as per procedures that were approved by Jeonbuk National University, Jeonju, Korea. The approval number was CBNU 2021-0115. Animals were anesthetized with isoflurane 24 h after the final challenge. Whole blood samples from mice were obtained by retro-orbital bleeding. All possible steps were taken to reduce the number of animals used in the study and alleviate their suffering.

### 2.4. Group Size, Randomization, and Blinding

Experimental mice were procured from the eligible pool of animals and randomly divided into six groups (n = 6 mice/group). The subjects were assigned to either a control group, where the mice had a sham induction using a saline challenge, or an experimental group, where the mice were induced with AR and treated with saline/CSE/dexamethasone (Dexa). The researchers that were responsible for the care of these animals had no knowledge about the pharmaceutical interventions that were administered to each experimental group.

### 2.5. Experimental Design

The mice used in this study were acclimated for a period of one week prior to their inclusion in the investigation. Six-week-old mice from the eligible pool of animals were randomly divided into six groups (6 mice in each group): (1) Group 1, control mice that received saline; (2) Group 2, AR mice that were sensitized with OVA and challenged with OVA; (3) Group 3, AR mice that received 50 mg/kg of CSE (AR + CSE-50); (4) Group 4, AR mice that received 100 mg/kg of CSE (AR + CSE-100); (5) Group 5, AR mice that received 200 mg/kg of CSE (AR + CSE-200); and (6) Group 6, AR mice that received 2.5 mg/kg of Dexa (AR + Dexa). Dexa was purchased from Sigma, USA (#D4902, Sigma, St. Louis, MO, USA), and the mice that were administered Dexa were used as a positive control group [37,38]. The administration of test compounds was carried out via oral gavage. All mice were sensitized with an intraperitoneal injection of 200 µL saline or 50 μg OVA (#A5503, Sigma, St. Louis, MO, USA) or 1 mg aluminum hydroxide (#77161, Thermo Fisher Scientific, Waltham, MA, USA) on days 1, 8, and 15. On days 22–28, mice were treated with an intranasal challenge with 20 µL OVA solution (10 mg/mL), aside from mice in the control group, who received saline. Moreover, from days 16–28, mice were administered either CSE (in the experimental groups) or Dexa (in the positive control group) 1 h prior to intranasal challenge via oral gavage daily. On day 29, shortly after OVA challenge, rubbing and sneezing frequency was observed in each mouse for 15 min. All the animals were euthanized the day following the final OVA challenge. 

### 2.6. Collection of Serum and Nasal Lavage Fluid (NLF)

Animals were anesthetized with isoflurane 24 h after the final challenge. Whole blood samples from mice were obtained via retro-orbital bleeding (i.e., from the eye sockets). Collected blood samples were immediately centrifuged at 10,000 RPM/min at 4 °C to obtain serum samples. NLF was collected using an 18-gauge catheter following a previously described procedure [39]. Briefly, the trachea was partially resected, after which a catheter was inserted to support the infusion of 1 mL saline through the nasal passages. The collected NLF sample was subjected to centrifugation to separate the supernatant from the cell pellet. ELISA was performed using the supernatant, while the total cell counts were enumerated by suspending the cell pellet in saline. Differential cell counts were estimated using cytospin. Diff-Quik staining was carried out using a Diff-Quik staining kit (Sysmex Co., Kobe, Japan) as per the manufacturer guidelines.

### 2.7. Histological Analysis

All histological analyses were performed as reported earlier [40]. Histopathological evaluations were performed to determine the influence of CSE on AR using head and lung tissues. First, head tissues were decalcified (decalcifying solution, HS-105, National Diagnostics, Atlanta, GA, USA) and processed as suggested previously [40]. Tissues were dissected, fixed, embedded, and sectioned (4.5 µm). Further, tissue sections were stained with hematoxylin and eosin (H&E, #517–28-2, Sigma, St. Louis, MO, USA), periodic acid Schiff (PAS, #ab245886, Abcam, Cambridge, UK), Giemsa (#G5637, Sigma, St. Louis, MO, USA), and Toluidine blue (TB, T3260, Sigma, St. Louis, MO, USA) to assess morphology, goblet cell hyperplasia, and mast cell infiltration.

### 2.8. Immunohistochemistry

All immunohistochemistry processes were performed as described previously [40]. Briefly, immunohistochemistry was conducted using a mouse-specific HRP/DAB (ABC) Kit with growth stimulation-expressed gene 2 (ST2) antibody (MBS7604494, Mybiosource, San Diego, CA, USA) and E-cadherin (#14472, Cell Signaling Technology, Danvers, MA, USA). Tissue sections were boiled and incubated at 4 °C with anti-ST2 and E-cadherin antibodies with a blocking solution. Sections were then exposed to biotinylated goat anti-rabbit IgG secondary antibody and DAB solution (Millipore, Billerica, MA, USA) was used for antigen labeling.

### 2.9. Determination of Cytokines and Immunoglobulins

The levels of interleukin (IL)-4 (#M4000B, R&D Systems, Inc., Minneapolis, MN, USA), IL-5 (#M5000, R&D Systems, Inc., Minneapolis, MN, USA), IL-12 (#M1270, R&D Systems, Inc., Minneapolis, MN, USA), IL-13 (#CSB-E04602m, CUSABIO, Houston, TX, USA), interferon-gamma (#IFN-γ, MIF00, R&D Systems, Inc., Minneapolis, MN, USA), occludin (#MBS729514, Mybiosource, San Diego, CA, USA), and ZO-1 (#MBS2602867, Mybiosource, San Diego, CA, USA) in NLF were assessed using commercially available ELISA kits. Similarly, ELISA kits were used to quantify Th2 cells associated with immunoglobulins like OVA-specific IgE (#439807, BioLegend, Inc., San Diego, CA, USA) as well as anti-ovalbumin IgG1 (#500830, Cayman, Ann Arbor, MI, USA) and IgG2a (#3015, Chondrex, Woodinville, WA, USA) in serum.

### 2.10. Immunoblotting

Immunoblotting was carried out in accordance with the procedures described in previous reports [41]. The collected tissues were lysed, separated, and transferred onto PVDF membranes. Next, they were blocked with 5% skimmed milk and probed with primary antibodies against IL-33 (1:1000, #88513, Cell Signaling, Danvers, MA, USA), anti-rabbit IgG (1:1000, #7074, Cell Signaling, Danvers, MA, USA), and β-actin (1:1000, #4970, Cell Signaling, Danvers, MA, USA). The membranes were then incubated with species-specific secondary antibodies, and the resulting protein signals were developed using ECL solution and quantified by measuring the band intensity.

### 2.11. Statistical Analysis

All data and statistical analyses were conducted while adhering to the recommended guidelines [42]. The statistical calculations and associated analyses were performed using GraphPad Prism version 8.4 (GraphPad Software, San Diego, CA, USA). The data are presented as mean ± SEM. For multiple comparisons, Student’s t-test and one-way ANOVA with Dunnett’s test were used. The significance level was set at *p* < 0.05.

## 3. Results

### 3.1. Active Components of CSE 

In total, six compounds were identified. The bioactive compounds in CSE are procyanidin B2, procyanidin C1/C2, rutin, baicalin/apigenin 7-glucuronide, diosmetin 7-glucuronide/6-o-methylscutellarin/diosmetin 3-glucuronide, and a-amylcinnamyl formate (Figure 1).

### 3.2. CSE Inhibits Nasal Symptoms of AR and Suppresses Immunoglobulin Levels in AR Mice

To determine the potential anti-allergic effect of CSE, behaviors such as nasal rubbing and sneezing were monitored among OVA-induced AR mice for 15 min shortly after OVA challenge on day 29. Rubbing and sneezing could be indicative of various underlying conditions or reactions; however, they are most commonly associated with allergic responses and respiratory tract irritants. These behaviors validate the induction of AR and confirm the overall success of the model. The observations for the present study revealed that the mice in the control and Dexa groups experienced significantly less sneezing and rubbing than those in the AR groups. By contrast, the administration of CSE dose-dependently reduced the occurrences of sneezing and rubbing. Moreover, these incidences were significantly lower in the CSE-treated group than they were in the AR group (Figure 2B,C).

Immunoglobulins are crucial to the development of AR. Immunoglobulins can bind to mast cells in nasal and respiratory tissues and cause them to release inflammatory mediators, thereby inducing AR symptoms. To investigate the effects of CSE on systemic allergic responses, the levels of OVA-specific IgE, IgG1, and IgG2a in OVA-induced AR mice were quantified. OVA-specific IgE and OVA-specific IgG1 levels were found to be significantly higher in mice in the AR groups than those in the control group. However, CSE dose-dependently downregulated IgE and IgG1 (Figure 2D,E). Further, CSE administration upregulated OVA-specific IgG2a in a dose-dependent manner (Figure 2F). Taken together, these findings suggest that CSE has the ability to regulate immunoglobulin levels in AR mice.

### 3.3. CSE Reduces Infiltration of Inflammatory Cells in the NLF of AR Mice

In AR, hyperresponsiveness of the immune system to allergens can trigger immune cells to release inflammatory stimulators, thus resulting in sneezing and itching. Activation of the inflammatory response is a crucial factor in the progression of AR. To determine the anti-inflammatory effects of CSE on AR, inflammatory cells in NLF were measured. The findings indicated a significant rise in the overall count of inflammatory cells in the AR group compared to the other groups. However, the administration of CSE dose-dependently reduced the number of inflammatory cells. Specifically, the numbers of inflammatory cells in groups treated with 100 or 200 mg/kg of CSE were significantly lower than those in the AR groups (Figure 3B,C). Further, a comparative analysis of differential *cell observations*—including epithelial cells, macrophages, eosinophils, neutrophils, and lymphocytes—demonstrated that the AR group had a higher abundance of these cells compared to the other groups. Moreover, CSE dose-dependently reduced the numbers of epithelial cells, macrophages, eosinophils, neutrophils, and lymphocytes (Figure 3C). These observations suggest that CSE supplementation reduces the total count of inflammatory cells in the NLF, thus highlighting the anti-inflammatory properties of CSE against AR.

### 3.4. CSE Prevents Pathological Alterations in Nasal Mucosa and Lung Tissues of AR Mice

Multiple staining methods were employed to detect pathological changes in the nasal mucosa and lung tissues. Inflammation within the nasal cavity is a hallmark of AR, where hyperresponsiveness triggers an inflammatory response in nasal passages. Mast cells can release histamine, thus causing blood vessels to dilate and nasal tissues to swell. Goblet cells produce excess mucus, leading to congestion and runny nose. Eosinophils and other immune cells can infiltrate the area, thus prolonging inflammation. Aberrant morphology in the nasal mucosa was observed in OVA-induced AR mice, as evidenced by the invasion of numerous inflammatory cells along with the swelling of the epithelial layer in the nasal septum. Consequently, AR mice exhibited more nasal septum thickening compared to control group mice. Interestingly, treatment with CSE 200 mg/kg significantly ameliorated both the accumulation of inflammatory cells and the swelling of the epithelial layer (Figure 4A,B). Rhinorrhea, which is a common nasal allergic symptom in AR, is characterized by runny or watery nasal discharge. Mucus production by goblet cells plays a critical role in this condition. PAS staining of tissues revealed the presence of goblet hyperplasia in OVA-induced AR mice, while goblet cell hyperplasia was alleviated after CSE treatment (Figure 4A,C). Further, Giemsa staining showed the major penetration of eosinophils into sub-mucosal tissues of OVA-induced AR mice. By contrast, CSE/Dexa treatment reduced the penetration of eosinophils (Figure 4A,D). TB staining revealed the presence of mast cells. Here, mast cells were found to be more highly recruited into the nasal tissues of OVA-induced AR mice compared to those in the control group (Figure 4A,E). As was expected, CSE treatment effectively reduced the number of infiltrated mast cells in subepithelial and epithelial layers.

### 3.5. Influence of CSE on Inflammation in Lung Tissues of OVA-Induced AR Mice

In the nasal tissues of AR mice, the submucosa exhibited thickening due to infiltration of inflammatory cells. The epithelial layer showed swelling while the goblet cells within the epithelium displayed excessive mucus production. To further determine the influence of CSE on OVA-induced AR, lung tissues were stained with H&E and PAS. Similar inflammatory features, such as increased infiltration of inflammatory cells and goblet cell hyperplasia, were observed in the lung tissues of AR mice. H&E staining revealed an inflammatory feature in the lung tissues of AR mice, where edema and thickened bronchial epithelial cells were observed. Interestingly, the morphology of lung tissue in mice administered with CSE or Dexa was comparable to that of mice in the control group. Moreover, a dense epithelial layer and greater goblet cell hyperplasia were observed in the OVA-induced AR group, while those in CSE-treated groups showed a thinner epithelial layer along with reduced goblet cell hyperplasia (Figure 5). These findings further confirm the anti-inflammatory effect of CSE in the OVA-induced mouse model.

### 3.6. CSE Inhibits IL-33 and ST2 in OVA-Induced AR Mice

The production and release of suppression of tumorigenicity-2 (ST2) in the alveolar epithelium are strongly associated with the development of inflammation in pulmonary diseases [43]. Previous reports have suggested that the IL-33-ST2 (IL-1RL1) axis is the root cause of asthma [44,45,46]. Moreover, IL-33, which is an epithelium-derived cytokine, has been shown to play an essential role in coupling the innate and adaptive immune responses mediated by Th2 cytokines [47]. Thus, the effects of CSE on IL-33 and ST2 protein levels in OVA-induced AR mice were determined. ST2, the receptor for IL-33, is generally predominantly localized on the epithelial cells and inflammatory cell membranes of mice. Here, immunohistochemistry observations on OVA-induced AR mice revealed a significant increase in ST2 positive area, while CSE administration was shown to effectively regulate ST2 distribution (Figure 6C,D). Moreover, immunoblot analysis revealed that the IL-33 and ST2 levels were significantly inhibited in the CSE- or Dexa-treated groups compared to the corresponding levels in OVA-challenged AR mice (Figure 6A,B; Appendix A).

### 3.7. CSE Modulates E-Cadherin, Occludin, and Zonula Occludens (ZO)-1 Expression in AR Mice

Previous reports suggest that inflammation-induced oxidative stress could hinder the function of the epidermal barrier by damaging the epidermal keratinocytes [48]. Specifically, E-cadherin plays a role in cell−cell adhesion, while occludin and ZO-1 are involved in maintaining tight junctions, which is crucial for barrier function [49,50]. Therefore, the present study measured the levels of E-cadherin, ZO-1, and occludin to determine the influence of CSE on epidermal barrier function. The immunohistochemistry analysis demonstrated that E-cadherin expression was significantly elevated upon treatment with 200 mg/kg of CSE compared to the corresponding level in the OVA-induced AR group (Figure 7D,E). Moreover, the expression levels of E-cadherin, occludin, and ZO-1 were significantly higher in CSE− or Dexa− treated mice than they were in OVA-induced AR mice (Figure 7A–C). Collectively, these observations demonstrate the beneficial effects of CSE administration on the recovery of airway epithelial barrier disruption.

### 3.8. CSE Regulates T Helper 2 (Th2) and Th2 Associated Cytokine Production in NLF of AR Mice

AR is primarily associated with an imbalance in the level of Th2 cytokines, which play a critical role in producing immunoglobulins and recruiting eosinophils. Conversely, Th1 cytokines typically counterbalance the Th2 response. They are often downregulated in AR. In this specific test, the levels of IFN-γ and IL-12 in NLF were found to be considerably lower in the OVA-induced AR group than they were in the control group. However, treatments with CSE enhanced the levels of IFN-γ and IL-12 in the NLF in a dose-dependent manner (Figure 8A,B). By contrast, the levels of Th2-associated cytokines such as IL-4, IL-5, and IL-13 in NLF were significantly upregulated in OVA-induced AR mice compared to those in the control group. However, the concentrations of these cytokines decreased substantially upon CSE supplementation (Figure 8C–E).

## 4. Discussion

AR is becoming a growing global health concern due to a significant increase in reported cases over the past few years. Its rising prevalence can be attributed to a number of factors such as urbanization, climate change, and increased allergen exposure. Specifically, emerging pollutants such as nanoparticles have been indicated to pose a heightened risk to human health as they have the capacity to infiltrate deeply into the respiratory system, thereby potentially exacerbating AR. AR is often linked to asthma, rhinosinusitis, and allergic conjunctivitis. At present, it is typically managed using steroidal and nonsteroidal anti-inflammatory agents. However, side effects linked to the use of steroidal and nonsteroidal anti-inflammatory agents have prompted increased consideration of the use of functional constituents that can promote health without side effects. Thus, in the current study, we evaluated the potential effect of CSE on AR in an OVA-induced AR mouse model. We also extrapolated the potential mechanism underlying the beneficial effect of CSE on AR. CSE is a polyphenol-rich product. It contains anti-inflammatory and antioxidant compounds such as quercetin, kaempferol, and various flavonoids [51]. Moreover, most of the positive implications of CSE can be attributed to the presence of procyanidins and other polyphenols like rutin and baicalin/apigenin 7-glucuronide; the UPLC data suggest that CSE contains significant quantities of these bioactive compounds. In particular, compounds like rutin are recognized for their ability to defend against lung injury [52,53], while procyanidins are known to aid in preventing inflammation in the airways [54,55]. Further, procyanidin can enhance vascular health, which in turn helps optimize the transportation of oxygen and reduce damage caused by inflammation [56]. These anti-inflammatory and antioxidant properties all play crucial roles in regulating respiratory diseases.

In this study, CSE supplementation was found to greatly reduce symptoms such as rubbing and sneezing. These observations aligned with the findings of previous reports showing that polyphenols could alleviate nasal symptoms [57,58]. Airway inflammation is a crucial factor in the development of asthma and AR, as it can cause structural alterations in the airway via the infiltration of inflammatory cells [59]. Moreover, in line with previous studies [60], our findings demonstrated that OVA challenge could significantly exacerbate airway inflammation and pathophysiology in AR mice. Nevertheless, CSE treatments significantly reduced airway inflammation in AR mice. Moreover, CSE at 200 mg/kg appeared to be equally effective as Dexa 2.5 mg/kg in reducing the infiltration of inflammatory cells into the NLF. The activation of an AR trigger leads to the infiltration of numerous inflammatory cells into the nasal tissue, which in turn results in inflammation within the airway and nose. Further, histopathological observations indicated alterations in eosinophil infiltration, mast cell count, and airway epithelial thickness. However, the administration of CSE lowered the epithelial swelling in the nasal mucosa and lung tissues, and it also regulated the excessive production of mucus-secreting goblet cells (Figure 3 and Figure 4). These observations suggest that CSE has a positive influence on pathological injury. AR is well known to involve a complex interplay of immune cells, including eosinophils, macrophages, and lymphocytes. Eosinophils are central players, as they release inflammatory mediators upon encountering allergens [61,62], thus leading to tissue damage and symptoms such as congestion and itching. Macrophages can act as sentinels, detecting allergens and initiating immune responses [63,64], and ultimately exacerbating inflammation. Meanwhile lymphocytes—specifically, T helper cells—can orchestrate immune responses, thus promoting IgE production and eosinophil activation. Notably, allergic lung inflammation can be induced by direct interactions between macrophages and mucosal epithelial cells. Here, we observed that OVA sensitization enhanced the activation of eosinophils and macrophages while CSE suppressed OVA-exacerbated trafficking. This specific observation effectively eliminates the possibility that a chemotactic gradient is aided by bronchial epithelial cells in the airway [65].

Immunoglobulins, specifically IgE antibodies, play pivotal roles in the development of AR. It has also been established that there is a substantial correlation between antigen-specific IgE and IgG1 in allergic mouse models [66]. IgE antibodies can adhere to mast cells and basophils scattered throughout the respiratory tract. Therefore, when exposed to the same allergen, IgE antibodies on mast cells and basophils can recognize and bind to the allergen, thus triggering the release of inflammatory stimulators. Therefore, immunoglobulins are identified as indicators of Th2 cells, while IgG2a acts as a marker for Th1 cells [67]. Here, CSE reduced OVA-specific IgE and IgG1 while increasing IgG2a. Consequently, an increased IgG2a/IgG1 ratio is anticipated with the supplement of CSE, which is indicative of a protective immune response in allergic inflammation diseases [68]. The IgG2a/IgG1 ratio indicates a balance between Th1 and Th2 immune responses, where a higher ratio suggests reduced allergic inflammation. In this condition, there could be a greater presence of anti-inflammatory Th1 responses compared to pro-inflammatory Th2 responses [69,70]. The interaction between IgE, mast cells, and the mucosal layer is critical to allergic responses. It can result in characteristic symptoms of airway inflammation, including bronchoconstriction, coughing, wheezing, and nasal congestion. Understanding this relationship is key to developing targeted therapies to mitigate airway inflammation in conditions such as AR and asthma. The histopathological analyses conducted and immunoglobulin measurements taken in the present investigation strongly indicate that CSE has a positive effect on OVA-induced AR.

The role played by epithelial cells in the development of AR has attracted increasing attention in recent years, due to their critical contribution to immunoregulatory processes that govern both innate and specific immunity [71]. Previous reports have indicated that, by activating the ERK1/2 pathway in AR, the IL-33/ST2 axis can stimulate inflammatory responses in nasal mucosal epithelial cells [72]. In general, IL-33 is typically disseminated into extracellular space after being swiftly released from the nucleus of necrotic cells. Further, in response to injury, IL-33 binds to a membrane-bound ST2 receptor via its cytokine domain [73]. There is also a strong correlation between the production and release of ST2 in the alveolar epithelium and the onset of inflammation in pulmonary disorders [43]. Previous reports have suggested that the IL-33-ST2 (IL-1RL1) axis is the root cause of asthma [44,45,46]. Further, IL-33 is crucial in coupling the innate and adaptive immune responses mediated by Th2 cytokines [47]. In line with these observations, the immunohistochemistry observations of the present study indicated a significant increase in ST2 positive area, while CSE administration effectively regulated ST2 distribution (Figure 6A,B). Further, CSE significantly inhibited IL-33 levels in AR mice. Taken together, these observations strongly indicate that nasal epithelial cells can mediate cellular inflammation through the IL-33/ST2 axis.

IL-33 can impact the expression and function of various tight junction proteins, including E-cadherin, occludin, and ZO-1 [74], which are essential for maintaining the integrity of the airway epithelial barrier. IL-33 can downregulate the expression of E-cadherin, occludin, and ZO-1 in airway epithelial cells [75,76]. This downregulation weakens the connections between adjacent epithelial cells, thus making the barrier less effective in preventing the passage of allergens, pathogens, and other irritants from the airway lumen into underlying tissues. Moreover, reduced levels of E-cadherin, occludin, and ZO-1 can lead to increased permeability of the airway epithelium. This increased permeability allows allergens, inflammatory mediators, and immune cells to easily breach the barrier, thereby leading to heightened inflammation and allergic responses. This barrier disruption is a crucial aspect of airway inflammation, which contributes to the pathogenesis of AR. Here, the results of immunohistochemistry analysis revealed that E-cadherin expression was significantly elevated upon treatment with 200 mg/kg of CSE in OVA-induced AR mice. CSE also facilitated the recovery of E-cadherin, occludin and ZO-1 in AR mice (Figure 7). Altogether, these observations lead us to believe that CSE can positively regulate the disruption of the airway epithelial barrier during pathological injury caused by AR.

IFN-γ is known to be a principal Th1 effector cytokine that can inhibit naive T cell differentiation into Th2 cells and trigger macrophage production [77]. IFN-γ also plays a vital role in regulating IgE antibodies [78,79]. Moreover, the overproduction of IgE can potentially increase allergic sensitivity and exacerbate symptoms. IFN-γ can also suppress eosinophil activation and recruitment, which are common in allergic inflammation and AR. This can have an anti-inflammatory effect by reducing eosinophil-mediated tissue damage. In the present study, CSE was shown to enhance IFN-γ levels in a dose-dependent manner (Figure 8A). The observation of higher IFN-γ levels in the CSE group demonstrate its ability to switch Th2 to Th1. This shift can help counteract allergic responses, reduce inflammation, and potentially alleviate symptoms. Further, Th2 cells release cytokines that can activate B cells and produce IgE specific to allergens, thus resulting in the manifestation of severe symptoms. IL-4 is also recognized for its ability to increase the synthesis of IgE, while IL-5 is responsible for promoting the proliferation and differentiation of eosinophils, and IL-13 plays a key role in promoting the production of mucus and airway hyperreactivity [80,81]. Since CSE administration was found to reduce IL-4, IL-5, and IL-13 levels (Figure 8C–E), it could be concluded that CSE influences IgE synthesis, mucus production, and eosinophilic growth and differentiation. IL-12 plays a crucial role in modulating airway inflammation, primarily by promoting a Th1 immune response. This function of IL-12 might also influence the release of IFN-γ. IL-12 also plays a major role in activating cytotoxic T cells, which are crucial for controlling infections and suppressing excessive immune responses. Consistent with these findings, the present study also noted that IL-12 was downregulated in OVA-induced AR whereas CSE dose-dependently restored IL-12 levels. Collectively, these results suggest that the IL-33/ST2 axis plays a crucial role in the pathogenesis of AR, and that the supplementation of CSE effectively regulated the IL-33/ST2/Th2 signaling pathway in nasal epithelial cells with the outcome of exerting a protective effect against AR (Figure 9).

The findings of the current study on mice are crucial for translating CSE into a functional food that prevents airway inflammation in humans. As CSE is a plant extract, there is a need for additional preclinical investigations using superior animal models to bridge the gap between animal and human studies. The current study has also provided a clear understanding of the mechanism of action, which is expected to be advantageous in expediting the development of clinical trial protocols for CSE in the near future. Moreover, collaborative efforts between researchers, clinicians, and nutritionists could further help pave the way for the development of CSE as a functional food.

## 5. Conclusions

AR is a global health concern that involves airway hyperresponsiveness, mucus hypersecretion, and eosinophilic infiltration in the airways. This study suggests that CSE can alleviate the progression of OVA-induced AR in mice by regulating the infiltration of inflammatory cells, accumulation of mucus, and OVA-specific immunoglobulins. These multiple types of interference by CSE modulate the IL-33/ST2/Th2 signaling pathway. This is attributable to the presence of active compounds such as rutin and procyanidins in CSE that can effectively regulate airway inflammation-associated disorders. Thus, interference with CSE may offer a new therapeutic approach for AR.

## Figures and Tables

**Figure 1 foods-13-00611-f001:**
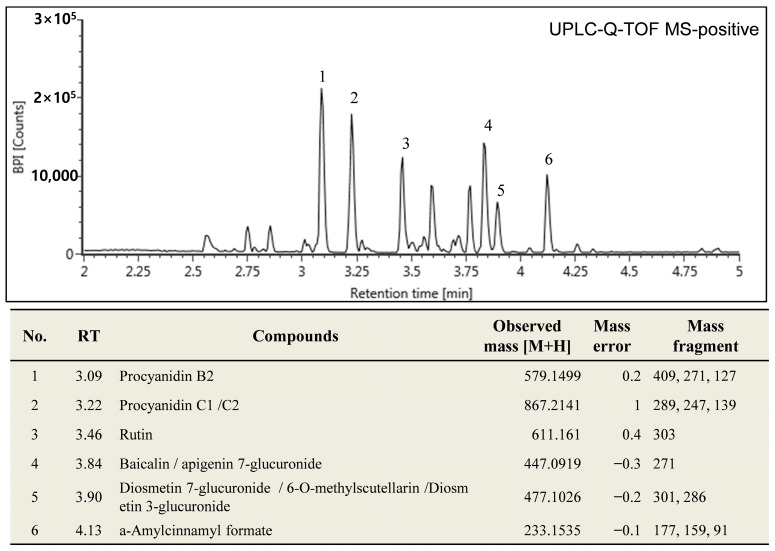
Chromatogram showing different components of CSE. Six compounds were identified in total. The bioactive compounds in CSE are Procyanidin B2, Procyanidin C1/C2, Rutin, Baicalin/apigenin 7-glucuronide, Diosmetin 7-glucuronide/6-O-methylscutellarin/Diosmetin 3-glucuronide, and a-Amylcinnamyl formate.

**Figure 2 foods-13-00611-f002:**
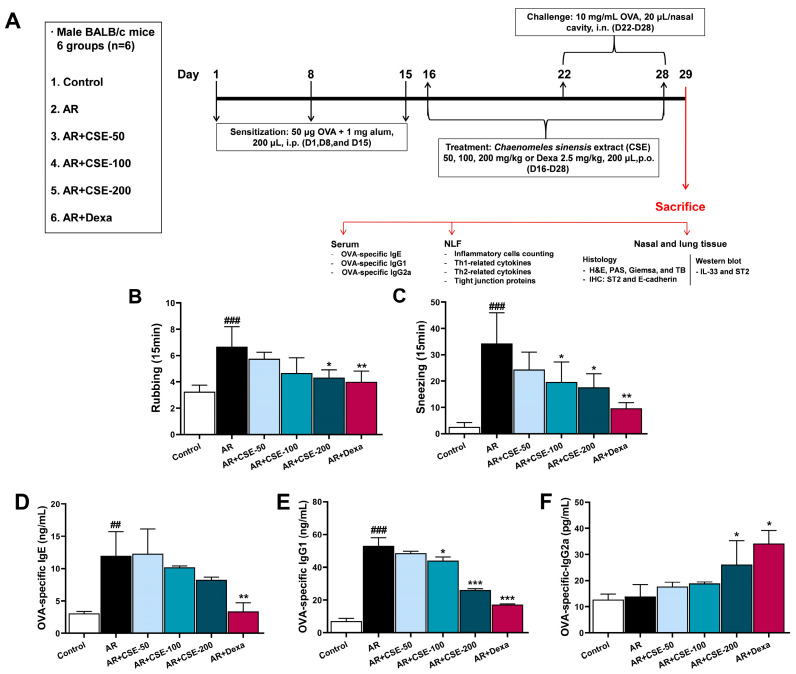
Experimental flow and CSE effect of reducing nasal symptoms and allergic responses via the regulation of immunoglobulin levels in an OVA-induced AR mouse model. (**A**) Schematic representation of timeline of experimental design. Mice were sensitized on days 1, 8, and 15 and then challenged with OVA via a nebulizer on days 22, 23, and 24. On days 25–31, mice were again challenged intranasally with OVA. Between day 16 and the end of the experiment, test compound (CSE or Dexa) was administered 1 h prior to intranasal challenge via oral gavage. Mice were administered with saline, Dexa (2.5 mg/kg), or CSE (50, 100, 200 mg/kg) once a day for 13 days. (**B**) Rubbing and (**C**) sneezing frequencies. (**D**) OVA-specific IgE, (**E**) OVA-specific IgG1, and (**F**) OVA-specific IgG2a levels in serum. Data are presented as mean ± SEM (n = 6/group). Significant differences at ^###^ *p* < 0.001 and ^##^ *p* < 0.01 vs. control group, *** *p* < 0.001, ** *p* < 0.01, and * *p* < 0.05 vs. AR group. OVA, ovalbumin; AR, allergic rhinitis; CSE, *Chaenomeles sinensis* extract; Dexa, dexamethasone; Control, mice that received saline; AR, mice that were sensitized with OVA and challenged with OVA; AR + CSE-50, AR mice that received 50 mg/kg of CSE; AR + CSE-100, AR mice that received 100 mg/kg of CSE; AR + CSE-200, AR mice that received 200 mg/kg of CSE; AR + Dexa, AR mice that received 2.5 mg/kg of Dexa.

**Figure 3 foods-13-00611-f003:**
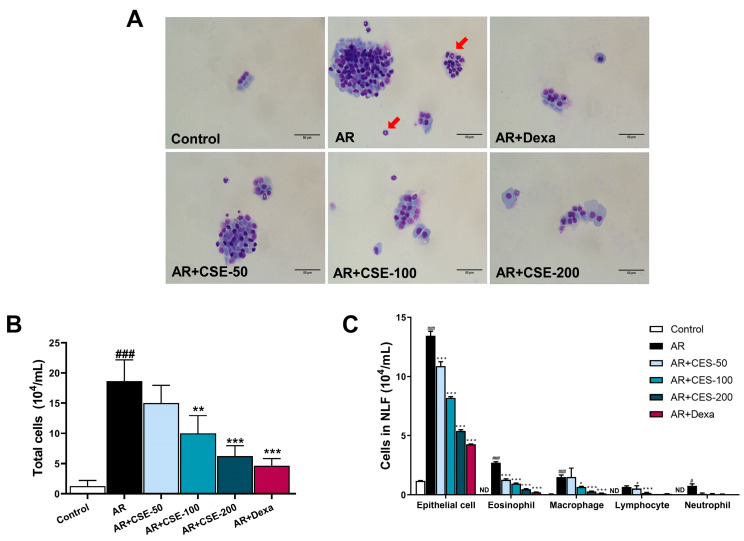
CSE reduces infiltration of differential inflammatory cells in NLF. (**A**) Diff-Quik-stained images. Red arrows indicate eosinophils. (**B**) Total counts of inflammatory cells. (**C**) Differential cell counts in NLF. ND: none detected. Data are presented as mean ± SEM (n = 6/group). Significant differences at **^###^** *p* < 0.001 and **^#^** *p* < 0.05 vs. control group, *** *p* < 0.001, ** *p* < 0.01, and * *p* < 0.05 vs. AR group. OVA, ovalbumin; AR, allergic rhinitis; CSE, *Chaenomeles sinensis* extract; Dexa, dexamethasone; Control, mice that received saline; AR, mice that were sensitized with OVA and challenged with OVA; AR + CSE-50, AR mice that received 50 mg/kg of CSE; AR + CSE-100, AR mice that received 100 mg/kg of CSE; AR + CSE-200, AR mice that received 200 mg/kg of CSE; AR + Dexa, AR mice that received 2.5 mg/kg of Dexa.

**Figure 4 foods-13-00611-f004:**
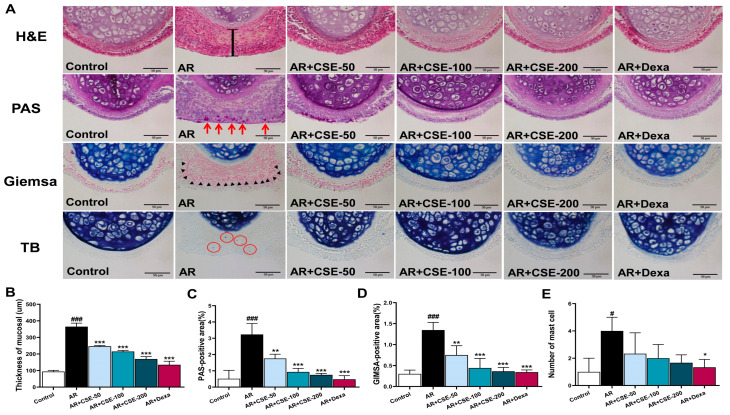
Histological analysis of AR nasal mucosa tissues to determine the influence of CSE on infiltration of inflammatory cells. (**A**,**B**) Representative H&E-stained images showing inflammatory characteristics and mucosal thickness. (**A**,**C**) Goblet cell quantification in PAS-stained nasal tissues. Red arrows indicate goblet cells. (**A**,**D**) Giemsa-stained images showing the infiltration of eosinophils. Black arrows indicate eosinophils. (**A**,**E**) Analysis of mast cells with TB-stained tissues (red circles indicate mast cells). Scale bar = 50 µm. Significant differences at ^###^ *p* < 0.001 and ^#^ *p* < 0.05 vs. control group, *** *p* < 0.001, ** *p* < 0.01, and * *p* < 0.05 vs. AR group. OVA, ovalbumin; AR, allergic rhinitis; CSE, *Chaenomeles sinensis* extract; Dexa, dexamethasone; Control, mice that received saline; AR, mice that were sensitized with OVA and challenged with OVA; AR + CSE-50, AR mice that received 50 mg/kg of CSE; AR + CSE-100, AR mice that received 100 mg/kg of CSE; AR + CSE-200, AR mice that received 200 mg/kg of CSE; AR + Dexa, AR mice that received 2.5 mg/kg of Dexa.

**Figure 5 foods-13-00611-f005:**
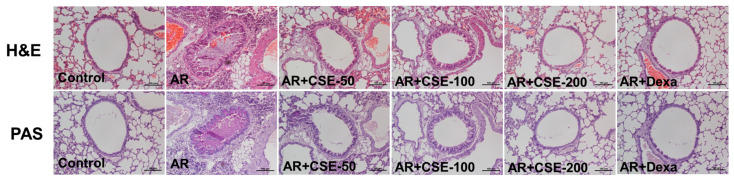
Histological analysis of lung tissue with representative H&E and PAS stain images of lung tissues. AR mice exhibited a variety of inflammatory features: epithelial swelling, excessive mucus secretion, and severe infiltration of inflammatory cells surrounding bronchus. Scale bar = 100 µm. OVA, ovalbumin; AR, allergic rhinitis; CSE, Chaenomeles sinensis extract; Dexa, dexamethasone; Control, mice that received saline; AR, mice that were sensitized with OVA and challenged with OVA; AR + CSE-50, AR mice that received 50 mg/kg of CSE; AR + CSE-100, AR mice that received 100 mg/kg of CSE; AR + CSE-200, AR mice that received 200 mg/kg of CSE; AR + Dexa, AR mice that received 2.5 mg/kg of Dexa.

**Figure 6 foods-13-00611-f006:**
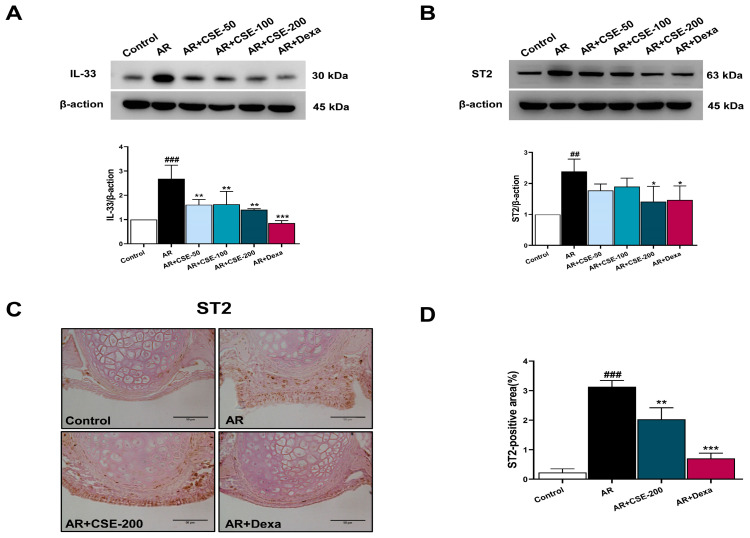
CSE regulates IL-33 and ST2 in AR mice. (**A**) Immunoblotting of IL-33 and respective quantification of IL-33. (**B**) Immunoblotting of ST2 and respective quantification of ST2. (**C**) Representative immunohistochemistry images showing the expression of ST2 in the epithelial layer of different groups. Scale bar = 50 µm. (**D**) Quantification of ST2 positive area. Positive area is expressed as a percentage. Data are presented as mean ± SEM (n = 6/group). Significant differences at **^###^** *p* < 0.001 and **^##^** *p* < 0.01 vs. control group, *** *p* < 0.001, ** *p* < 0.01, and * *p* < 0.05 vs. AR group. OVA, ovalbumin; AR, allergic rhinitis; CSE, *Chaenomeles sinensis* extract; Dexa, dexamethasone; Control, mice that received saline; AR, mice that were sensitized with OVA and challenged with OVA; AR + CSE-200, AR mice that received 200 mg/kg of CSE; AR + Dexa, AR mice that received 2.5 mg/kg of Dexa.

**Figure 7 foods-13-00611-f007:**
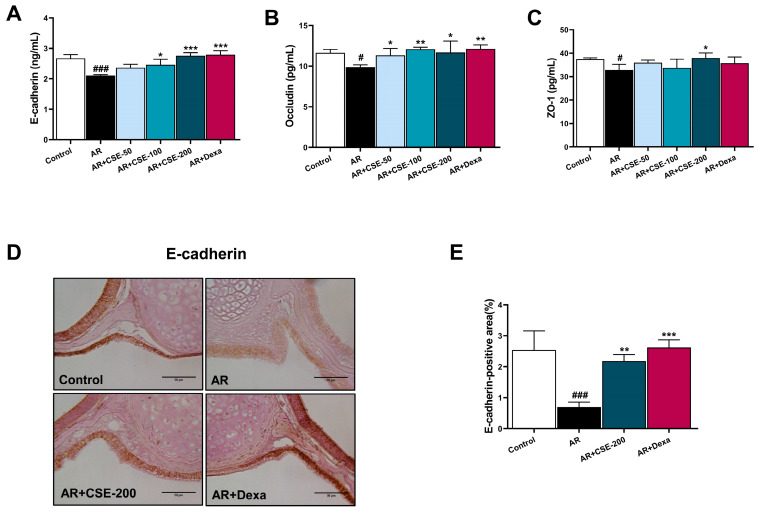
CSE modulates E-cadherin, Occludin, and ZO-1 expression in AR mice. Levels of (**A**) E-cadherin, (**B**) occludin, and (**C**) ZO-1 in NLF were determined using ELISA kits. (**D**) Representative immunohistochemistry images showing E-cadherin expression in the epithelium layer of different groups. Scale bar = 50 µm. (**E**) Quantification of E-cadherin positive area. Positive area is expressed as a percentage. Data are presented as mean ± SEM (n = 6/group). Significant differences at **^###^** *p* < 0.001, and **^#^** *p* < 0.05 vs. control group, *** *p* < 0.001, ** *p* < 0.01 and * *p* < 0.05 vs. AR group. OVA, ovalbumin; AR, allergic rhinitis; CSE, *Chaenomeles sinensis* extract; Dexa, dexamethasone; Control, mice that received saline; AR, mice that were sensitized with OVA and challenged with OVA; AR + CSE-50, AR mice that received 50 mg/kg of CSE; AR + CSE-100, AR mice that received 100 mg/kg of CSE; AR + CSE-200, AR mice that received 200 mg/kg of CSE; AR + Dexa, AR mice that received 2.5 mg/kg of Dexa.

**Figure 8 foods-13-00611-f008:**
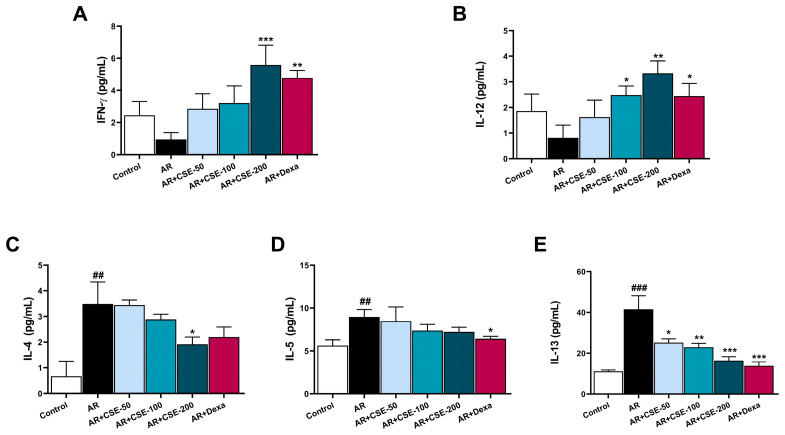
CSE ameliorates production of Th2 cytokines but increases Th1 levels in NLF. (**A**) IFN-γ, (**B**) IL-12, (**C**) IL-4, (**D**) IL-5, and (**E**) IL-13 levels were analyzed to determine the implications of CSE on OVA-induced AR mice. Data are presented as mean ± SEM (n = 6/group). Significant differences at **^###^** *p* < 0.001 and **^##^** *p* < 0.01 vs. control group, *** *p* < 0.001, ** *p* < 0.01, and * *p* < 0.05 vs. AR group. OVA, ovalbumin; AR, allergic rhinitis; CSE, *Chaenomeles sinensis* extract; Dexa, dexamethasone; Control, mice that received saline; AR, mice that were sensitized with OVA and challenged with OVA; AR + CSE-50, AR mice that received 50 mg/kg of CSE; AR + CSE-100, AR mice that received 100 mg/kg of CSE; AR + CSE-200, AR mice that received 200 mg/kg of CSE; AR + Dexa, AR mice that received 2.5 mg/kg of Dexa.

**Figure 9 foods-13-00611-f009:**
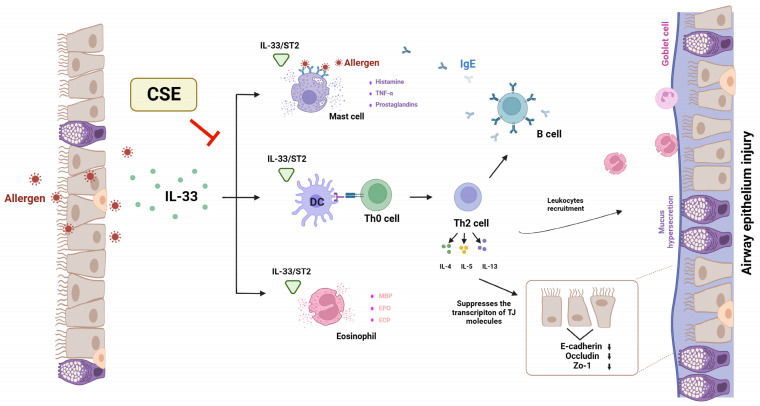
Potential molecular mechanism that drives the positive impact of CSE in OVA-induced AR mouse model. Allergen exposure stimulates the release of IL-33 from the nucleus of necrotic cells and its subsequent accumulation in extracellular space. Here, IL-33 interacts with membrane-bound ST2 receptor via its cytokine domain in response to cellular injury. This triggers Th2 immune responses and stimulates the differentiation of Th0 cells to Th2 cells. This cascade enhances the recruitment of immune cells such as eosinophils and mast cells to the airways. Moreover, Th2 cells suppress the transcription of tight junction proteins such as E-cadherin, occludin, and ZO-1. In conjunction with the cytokines secreted by Th2 cells, leukocytes are recruited and activated, thus resulting in allergic inflammatory symptoms. However, CSE can significantly inhibit IL-33 levels and subsequent ST2 in OVA-induced AR mice, indicating that CSE can provide a protective effect against OVA-induced AR by regulating the infiltration of inflammatory cells, accumulation of mucus, and OVA-specific immunoglobulins. These multiple forms of interference by CSE modulate the IL-33/ST2/Th2 signaling pathway.

## Data Availability

The original contributions presented in the study are included in the article/Appendix A, further inquiries can be directed to the corresponding author.

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
