# Peer review of "Chaenomeles sinensis Extract Ameliorates Ovalbumin-Induced Allergic Rhinitis by Inhibiting the IL-33/ST2 Axis and Regulating Epithelial Cell Dysfunction"

_foods, 2024, doi:10.3390/foods13040611_

Round 1

Reviewer 1 Report

Comments and Suggestions for Authors

The paper by Jin et al., examined Chaenomeles sinensis extract (CSE) as a bioactive herb to attenuate the symptoms associated with allergic rhinitis (AR) using a mouse model.  BALB/c mice were treated with varying doses of CSE along with ovalbumin to induce AR.  Various parameters of AR were assessed (e.g., cytokines; inflammatory cells; epithelial integrity).  CSE was effective in attenuating many of the adverse symptoms of AR and in some cases in a dose-dependent fashion.  The manuscript requires improvement in grammar/ language (e.g., line 41-42; "...10% and 30% ...").  Specific comments:

- was the CSE given every day?  Not clear.

- how was 6 mice per group determined?  Power test?

- a Dunnett's test everything is compared to one group - the Figures show all pair-wise comparisons

- what did the dexamethasone group represent?  A positive control?

- many of the Figures are much too small to adequately evaluate.

- the Discussion should include how these results translate to humans - how would humans received CSE and at what dose?  How were the doses for this study chosen?

Comments on the Quality of English Language

Needs some improvement

Author Response

REBUTTAL LETTER

We would like to express our sincere thanks to the reviewers who identified areas of our manuscript that needed corrections/formations/ modifications. We have responded to the reviewer’s comments point by point

Reviewer 1

The manuscript requires improvement in grammar/ language (e.g., line 41-42; "...10% and 30% ...").  Specific comments:

- was the CSE given every day?  Not clear.

Response: We thank the reviewer for careful observations. As suggested, we have corrected the minor error. Regarding CSE administration, Between days 16 and 28, mice were administered with CSE or Dexa via oral gavage daily. Revised text in the revised manuscript as follows,

Regarding grammatical errors

Typically, AR persisting throughout life, and impacts approximately 10% to 30% of the global population”

Regarding CSE administration

“All mice were sensitized with an intraperitoneal injection of 200 µL saline or 50 μg OVA (#A5503, Sigma, USA) or 1 mg aluminum hydroxide (#77161, Thermo Fisher Scientific, USA) on days 1, 8, and 15. On days 22-28, mice were treated with an intranasal challenge with 20 µL OVA solution (10 mg/mL), while mice in the control group received saline. Moreover, from day 16-28, mice were administered CSE or Dexa 1h prior to intranasal challenge via oral gavage daily. On day 29, shortly after OVA challenge, rubbing and sneezing frequency by each mouse was observed for 15 min. All the animals were euthanized day after the final OVA challenge. The experimental design is illustrated in Fig. 1A”

- how was 6 mice per group determined?  Power test?

Response: We appreciate reviewer for the valuable comment. In this study, we used 6 mice in each group. For a small sample size of 6, conventional normality tests such as the Shapiro-Wilk or Kolmogorov-Smirnov test are unsuitable due to their low statistical power. However, based on our experience,  we consider visual inspection methods like Q-Q and box plots to assess the normality. These graphical methods offer a rough indicator of whether data follows a normal distribution. We acknowledge that the small sample size may pose a significant limitation to the research. Still, these novel observations establish the fundamental basis for subsequent animal and clinical investigations.

- a Dunnett's test everything is compared to one group - the Figures show all pair-wise comparisons

Response: No, the comparisons between the categories were based on the significance of each group with respect to the specific tests. In certain instances, we have compared CSE groups to Dexa. In most cases, CSE groups were frequently compared to the AR control group.

- what did the dexamethasone group represent?  A positive control?

Response: Thank you for the comment. dexamethasone group was used as the positive control group.

- many of the Figures are much too small to adequately evaluate.

Response: Thank you for the suggestion. As indicated, we made attempt to improve the image size without affecting the resolution. We hope revised figures are large enough for evaluation.

- the Discussion should include how these results translate to humans - how would humans received CSE and at what dose?  How were the doses for this study chosen?

Response: Thank you for the valuable suggestions. As suggested, we have included a paragraph on development of CSE as a functional food in the revised manuscript. However, dose translation is not always a straightforward process due to variations between species. Here, variations in drug distribution, absorption, and metabolism can be substantial. The significance of pharmacokinetic modeling in the prediction of human dosage is widely acknowledged. Therefore, in order to evaluate safety and observe pharmacological effects, phase I trials are essential. At this stage, we are not able to predict the dose for clinical trials. Regarding its administration, we believe that the tablet form of CSE is more suitable for clinical trials.

The updated text on commented aspects in revised manuscript as follows,

“The findings of the current study on mice are crucial for translating CSE into a functional food that prevents airway inflammation in humans. CSE, being a plant extract, is unquestionably safe for human consumption. However, additional preclinical investigations using superior animal models are essential to bridge the gap between animal and human studies. In addition, the study has provided a clear understanding of the mechanism of action that will be advantageous in expediting the development of clinical trial protocols for CSE in the near future. Furthermore, collaborative efforts between researchers, clinicians, and nutritionists could pave the path for the development of CSE as a functional food”

Reviewer 2 Report

Comments and Suggestions for Authors

In this manuscript entitled “Chaenomeles sinensis extract ameliorates ovalbumin-induced allergic rhinitis by inhibiting the IL-33/ST2 axis and regulating epithelial cell dysfunction”, the authors report on the protective effect of Chaenomeles sinensis extract against ovalbumin-induced allergic rhinitis in mice. They investigated the influence of the extract, which is rich in bioactive compounds such as terpenoids, phenols and flavonoids, on nasal symptoms, inflammatory cells, OVA-specific immunoglobulins, cytokine production, and nasal histopathology in the OVA-induced AR mouse model.

The topic is very interesting and the results presented seem promising. The manuscript is well-structured, clearly written and the methodology used is appropriate. Only a few points should be revised:

-   Considering that some authors have analyzed and studied extracts from different parts of the plant (e.g. leaves in ref. 18), I suggest that the authors define which parts of the plant they have extracted. I assume they are talking about fruits, but these are not precisely defined anywhere.

-   In the "References" section: In many references, the name of the journal and the DOI number are omitted (e.g. in references 7, 14-16, 18, 20-23, 30, 35, 36, 40, 46, 48 and 52). In addition, the abbreviated name of the journal should be given in all cited references (in references 10, 19, 25-26, 39, 49, and 51 the full names of the journals are given). Also, reference 37 is not correctly cited and should be corrected in: Sedgwick, J.B.; Nagata M. Mechanism of eosinophil activation. In: Asthma and Rhinitis; Busse, W; Holgate, S., Eds.; Boston (MA): Blackwell Scientific; 2000; pp. 373–393.

-   References to recent studies on the plant extracts of Chaenomeles sinensis should be added:

1)      Yang, E.J.; Lee, S.H. Anti-Inflammatory Effects of Chaenomeles sinensis Extract in an ALS Animal Model.  Front  Biosci  2023, 28(12), 326. doi: 10.31083/j.fbl2812326.

2)      Xu, R.; Kuang, M.; Li, N. Phytochemistry and pharmacology of plants in the genus Chaenomeles. Arch Pharm Res 2023, 46, 825–854. doi:10.1007/s12272-023-01475-w

3)      Kang, J.S.; Zhao, X.Y.; Lee, J.H.; Lee, J-S.; Keum, Y-S. Ethanol Extract of Chaenomeles sinensis Inhibits the Development of Benign Prostatic Hyperplasia by Exhibiting Anti-oxidant and Anti-inflammatory Effects. J Cancer Prev 2022, 27(1), 42-49. doi: 10.15430/JCP.2022.27.1.42.

4)      Itoh, S.; Yamaguchi, M.; Shigeyama, K.; Sakaguchi, I. The Anti-Aging Potential of Extracts from Chaenomeles sinensis. Cosmetics 2019, 6(1), 21. doi:10.3390/cosmetics6010021

Following the above corrections and improvements, the manuscript is suitable for publication in Foods.

Author Response

REBUTTAL LETTER

We would like to express our sincere thanks to the reviewers who identified areas of our manuscript that needed corrections/formations/ modifications. We have responded to the reviewer’s comments point by point

Reviewer 2

In this manuscript entitled “Chaenomeles sinensis extract ameliorates ovalbumin-induced allergic rhinitis by inhibiting the IL-33/ST2 axis and regulating epithelial cell dysfunction”, the authors report on the protective effect of Chaenomeles sinensis extract against ovalbumin-induced allergic rhinitis in mice. They investigated the influence of the extract, which is rich in bioactive compounds such as terpenoids, phenols and flavonoids, on nasal symptoms, inflammatory cells, OVA-specific immunoglobulins, cytokine production, and nasal histopathology in the OVA-induced AR mouse model.

The topic is very interesting and the results presented seem promising. The manuscript is well-structured, clearly written and the methodology used is appropriate. Only a few points should be revised:

-   Considering that some authors have analyzed and studied extracts from different parts of the plant (e.g. leaves in ref. 18), I suggest that the authors define which parts of the plant they have extracted. I assume they are talking about fruits, but these are not precisely defined anywhere.

Response: Thank you for the valuable comment. In this study, we used fruits of the Chaenomeles sinensis for extraction process. As indicated, we have included the information in the revised manuscript. Updated text as follows,

Briefly, fruit of raw C. sinensis was dried, pulverized, and extracted with 20X of 50% ethanol at ± 50oC for 6 h. Next, CSE was filtered via cartridge filter. Following filtration, CSE was concentrated at 50oC under a vacuum with help of Rotavapor R-210 (BÜCHI Labortechnik AG, Flawil, Switzerland). Subsequently, CSE was freeze-dried and stored until use.

-   In the "References" section: In many references, the name of the journal and the DOI number are omitted (e.g. in references 7, 14-16, 18, 20-23, 30, 35, 36, 40, 46, 48 and 52). In addition, the abbreviated name of the journal should be given in all cited references (in references 10, 19, 25-26, 39, 49, and 51 the full names of the journals are given). Also, reference 37 is not correctly cited and should be corrected in: Sedgwick, J.B.; Nagata M. Mechanism of eosinophil activation. In: Asthma and Rhinitis; Busse, W; Holgate, S., Eds.; Boston (MA): Blackwell Scientific; 2000; pp. 373–393.

-   References to recent studies on the plant extracts of Chaenomeles sinensis should be added:

1)      Yang, E.J.; Lee, S.H. Anti-Inflammatory Effects of Chaenomeles sinensis Extract in an ALS Animal Model.  Front  Biosci  2023, 28(12), 326. doi: 10.31083/j.fbl2812326.

2)      Xu, R.; Kuang, M.; Li, N. Phytochemistry and pharmacology of plants in the genus ChaenomelesArch Pharm Res 2023, 46, 825–854. doi:10.1007/s12272-023-01475-w

3)      Kang, J.S.; Zhao, X.Y.; Lee, J.H.; Lee, J-S.; Keum, Y-S. Ethanol Extract of Chaenomeles sinensis Inhibits the Development of Benign Prostatic Hyperplasia by Exhibiting Anti-oxidant and Anti-inflammatory Effects. J Cancer Prev 2022, 27(1), 42-49. doi: 10.15430/JCP.2022.27.1.42.

4)      Itoh, S.; Yamaguchi, M.; Shigeyama, K.; Sakaguchi, I. The Anti-Aging Potential of Extracts from Chaenomeles sinensisCosmetics 2019, 6(1), 21. doi:10.3390/cosmetics6010021 

Response: Thank you for your valuable suggestion and information. Following understanding on mentioned references, we have included the relevant references in the revised manuscript. In the revised manuscript, reference 19, 20, and 34 belong to the recommended references.

Reviewer 3 Report

Comments and Suggestions for Authors

The manuscript entitled “Chaenomeles sinensis extract ameliorates ovalbumin-induced allergic rhinitis by inhibiting the IL-33/ST2 axis and regulating epithelial cell dysfunction” addresses the beneficial effect of Chaenomeles sinensis extract (CSE) in a mouse model of allergic rhinitis and the associated molecular mechanisms. Initially, the authors proved that CSE down-regulated OVA-specific IgE and lowered the Th1/Th2 cytokine ratio in nasal lavage. Moreover, CSE dampened the protein expression of IL-33/ST2, protected epithelial barrier integrity, and improved nasal and lung histopathological features. The current findings are interesting, and the manuscript is clearly written.

Comments:     

1) In the introduction section, the authors are advised to describe the ovalbumin-induced allergic rhinitis model, its advantages, and how it mimics the human condition.

2) In line 90, the authors stated “Animals were deeply anesthetized with diethyl ether”.

In fact, the American Veterinary Medical Association (AVMA) “Guidelines for the Euthanasia of Animals” (2013) has determined that the use of ether for euthanasia of animals is not acceptable. Ether is highly soluble, induces anesthesia slowly, and may be accompanied by agitation. Moreover, ether is irritating in animals, especially to the eyes and nose, and has been used to create a model for stress.

Please, double-check the name of the used anesthesia agent.

3) In section 2.4., how were the doses of CS extract selected? Please provide proper citations for the selected dose of CS extract.  Are the selected doses in rats relevant for human translation? Can you discuss the dose used for possible translation in humans, for example, by using conversion tables available in the literature using the Human effective dose (HED) formula= animal dose x animal Km/ human Km (Nair AB, Jacob S. A simple practice guide for dose conversion between animals and humans. J Basic Clin Pharm. 2016 Mar;7(2):27-31). I would suggest that authors address this point and add the answers/proper citations to section 2.4.

4) Likewise, please provide proper citations for the selected dose of dexamethasone.  

5) In immunohistochemistry (section 2.7), did the authors also perform a negative control to ensure the specific binding of antibody to target protein? Please, add the answer to section 2.7.

6) In the statistical analysis section, did the authors check data normality before proceeding to one-way ANOVA? Authors are advised to address this point and add the answers in the material and methods section

7) Since the study involves several experimental groups/treatments, statistical analysis is typically analyzed by ANOVA followed by a post-hoc test. The combination of t-test and one-way ANOVA might not be appropriate. Please revise the statistical comparisons accordingly.

8) In Figures, the panels shown are tiny and cannot be read by readers, for example, in Figures 1B, C, 3C, and 4B-E. Please, use bigger panels.

9) In Figure 6A,B, the authors are advised to describe the number of replicates used in Western blotting. Moreover, was the data extracted from independent samples?

10) The authors are advised to carefully revise the reference section. The authors are advised to unify the way they write the journal name. Sometimes it is written as an abbreviation (most references) while in other references it was written as a full name (reference 4). Please, follow the journal instructions in this regard.

Comments on the Quality of English Language

Minor editing of the English language is required.

Author Response

REBUTTAL LETTER

We would like to express our sincere thanks to the reviewers who identified areas of our manuscript that needed corrections/formations/ modifications. We have responded to the reviewer’s comments point by point

Reviewer 3

1) In the introduction section, the authors are advised to describe the ovalbumin-induced allergic rhinitis model, its advantages, and how it mimics the human condition.

Response: We appreciate the reviewer for thoughtful comment. Albumin from chicken eggs (OVA) was used to induce AR in inbred BALB/c mice. The mice model represents the most accepted approach to studying AR. Previous investigations employing OVA to induce AR have shown the presence of TH2 cytokines in the olfactory bulbs, prefrontal brain (PFC), temporal cortex, and hypothalamus. These regions contain multiple structures that have been proven to be sensitive to various forms of experimentally-induced allergies [1,2]. Furthermore, previous reports suggested the presence of anxiety-like behaviors in the initial stage of an allergic asthma model [3] [4] and offer evidence that comparable changes in behavior, as observed in humans with respiratory allergies, are similar to experimental induction of AR or asthma in rodents [1]. Therefore, we have confidence that OVA is highly effective in generating AR that closely resembles human conditions.

2) In line 90, the authors stated “Animals were deeply anesthetized with diethyl ether”.

In fact, the American Veterinary Medical Association (AVMA) “Guidelines for the Euthanasia of Animals” (2013) has determined that the use of ether for euthanasia of animals is not acceptable. Ether is highly soluble, induces anesthesia slowly, and may be accompanied by agitation. Moreover, ether is irritating in animals, especially to the eyes and nose, and has been used to create a model for stress.

Please, double-check the name of the used anesthesia agent.

 Response: We express our regret for the oversight in using the term diethyl ether. Currently, following the modification in regulations by AVMA, we have adopted the use of isoflurane as an anesthetic. The error has been rectified in the revised manuscript.

3) In section 2.4., how were the doses of CS extract selected? Please provide proper citations for the selected dose of CS extract.  Are the selected doses in rats relevant for human translation? Can you discuss the dose used for possible translation in humans, for example, by using conversion tables available in the literature using the Human effective dose (HED) formula= animal dose x animal Km/ human Km (Nair AB, Jacob S. A simple practice guide for dose conversion between animals and humans. J Basic Clin Pharm. 2016 Mar;7(2):27-31). I would suggest that authors address this point and add the answers/proper citations to section 2.4.

Response: We express our sincere thanks to the reviewer for their insightful feedback. Prior to determining the dosage of CSE, we conducted a comprehensive examination of relevant studies published in credible academic journals. Our literature review revealed that the majority of investigations using mice utilized doses of up to 200 mg/kg (irrespective of the target disease condition). Therefore, considering the prior studies, we selected a dosage range of 50-200 mg/kg for the study.

The list of references used for deciding the dose as follows,

KIM, J.-O., AN, G. & CHOI, J.-H. 2023. Protective effect of mixture of Acanthopanax sessiliflorum and Chaenomeles sinensis against ultraviolet B-induced photodamage in human fibroblast and hairless mouse. Food Science and Biotechnology.

YANG, E. J. & LEE, S. H. 2023. Anti-Inflammatory Effects of Chaenomeles sinensis Extract in an ALS Animal Model. 28.

ZHANG, R., ZHAN, S., LI, S., ZHU, Z., HE, J., LORENZO, J. M. & BARBA, F. J. 2018. Anti-hyperuricemic and nephroprotective effects of extracts from Chaenomeles sinensis (Thouin) Koehne in hyperuricemic mice. Food Funct, 9, 5778-5790.

Regarding translation of selected doses for human

Dose translation is not always a straightforward process due to variations between species. Here, variations in drug distribution, absorption, and metabolism can be substantial. The significance of pharmacokinetic modeling in the prediction of human dosage is widely acknowledged. Therefore, in order to evaluate safety and observe pharmacological effects, phase I trials are essential. At this stage, we are not able to predict the dose for clinical trials.

4) Likewise, please provide proper citations for the selected dose of dexamethasone.  

 Response: We have used DEXA as a positive control group that compared with the CSE treatment. As suggested, we have indicated the references which are relevant to the selection of dose of dexamethasone in revised manuscript. In the revised manuscript, reference 22 and 23 are relevant to the selection of dexamethasone dose.

5) In immunohistochemistry (section 2.7), did the authors also perform a negative control to ensure the specific binding of antibody to target protein? Please, add the answer to section 2.7.

Response: We thank the reviewer for careful observations. In this study, we have not used a negative control as study focuses on positive outcomes and we are confident on the methodologies that assure specificity.

6) In the statistical analysis section, did the authors check data normality before proceeding to one-way ANOVA? Authors are advised to address this point and add the answers in the material and methods section.

Response: In this study, we used 6 mice in each group. For a small sample size of 6, conventional normality tests such as the Shapiro-Wilk or Kolmogorov-Smirnov test are unsuitable due to their low statistical power. However, based on our experience, we consider visual inspection methods like Q-Q and box plots to assess the normality. These graphical methods offer a rough indicator of whether data follows a normal distribution. We acknowledge that the small sample size may pose a significant limitation to the research. Still, these novel observations establish the fundamental basis for subsequent animal and clinical investigations.

7) Since the study involves several experimental groups/treatments, statistical analysis is typically analyzed by ANOVA followed by a post-hoc test. The combination of t-test and one-way ANOVA might not be appropriate. Please revise the statistical comparisons accordingly.

Response: We thank the reviewer for careful observations. I used one-way ANOVA with Dunnett’s test were analyzed for AR and treatment groups. However, t-test analyzed the Control group and AR group.

8) In Figures, the panels shown are tiny and cannot be read by readers, for example, in Figures 1B, C, 3C, and 4B-E. Please, use bigger panels.

Response: Thank you for the suggestion. As indicated, we made attempt to improve the image size without affecting the resolution. We hope revised figures are large enough for evaluation.

9) In Figure 6A,B, the authors are advised to describe the number of replicates used in Western blotting. Moreover, was the data extracted from independent samples?

Response: Thank you for the comment. Immunoblot analysis was performed using independent samples in triplicates.

10) The authors are advised to carefully revise the reference section. The authors are advised to unify the way they write the journal name. Sometimes it is written as an abbreviation (most references) while in other references it was written as a full name (reference 4). Please, follow the journal instructions in this regard.

Response: Thank you for the comment. We have used the reference manager (ENDNOTE) for managing references. In the revised manuscript, we have corrected the errors in the reference section. We used MDPI style for arranging references.  

References

  1. Tonelli, L.H.; Katz, M.; Kovacsics, C.E.; Gould, T.D.; Joppy, B.; Hoshino, A.; Hoffman, G.; Komarow, H.; Postolache, T.T. Allergic rhinitis induces anxiety-like behavior and altered social interaction in rodents. Brain Behav Immun 2009, 23, 784-793, doi:10.1016/j.bbi.2009.02.017.
  2. Rosenkranz, M.A.; Busse, W.W.; Johnstone, T.; Swenson, C.A.; Crisafi, G.M.; Jackson, M.M.; Bosch, J.A.; Sheridan, J.F.; Davidson, R.J.J.P.o.t.N.A.o.S. Neural circuitry underlying the interaction between emotion and asthma symptom exacerbation. 2005, 102, 13319-13324.
  3. Holsboer, F.; Ising, M.J.E.j.o.p. Central CRH system in depression and anxiety—evidence from clinical studies with CRH1 receptor antagonists. 2008, 583, 350-357.
  4. Zoumakis, E.; Rice, K.; Gold, P.; Chrousos, G.J.A.o.t.N.Y.A.o.S. Potential Uses of Corticotropin‐Releasing Hormone Antagonists. 2006, 1083, 239-251.

Reviewer 4 Report

Comments and Suggestions for Authors

 Dear author, the manuscript presented has very relevant and novel information, however, I suggest the following aspects be taken into account

 Introduction section

 In this section it is suggested to integrate more information than the one indicated, among which the following stand out: 

 Line 65 it is indicated that C. sinensis contains bioactive compounds such as terpenoids and phenols, it is suggested to complete this information and include

 line 68 it is indicated that it inhibits oxidative stress, it is suggested to deepen this information.

 line 70 it is indicated that a richness in polyphenolic compounds so it is suggested to integrate more information of the compounds established for this mechanism.

 Results section

 This section provides a complete description of the studies carried out on the model and the effect of the different concentrations of the extract, covering different complementary techniques, and the results are clearly described. 

However, I consider a fundamental and important part to include is information on the phytochemical composition of C. sinensis from the background, as well as the characterization of the extracts obtained and tested in the animal model, in this way the results obtained can be discussed, although a discussion of the behavior of the different concentrations of the extracts is made, it is not clear which phytochemical compounds may be generating the effect.

Author Response

REBUTTAL LETTER

We would like to express our sincere thanks to the reviewers who identified areas of our manuscript that needed corrections/formations/ modifications. We have responded to the reviewer’s comments point by point

Reviewer 4

Dear author, the manuscript presented has very relevant and novel information, however, I suggest the following aspects be taken into account

 Introduction section

 In this section it is suggested to integrate more information than the one indicated, among which the following stand out: 

Line 65 it is indicated that C. sinensis contains bioactive compounds such as terpenoids and phenols, it is suggested to complete this information and include

Response: As suggested, we have included the essential information on these bioactive compounds relevant to respiratory distress. Updated text in the revised manuscript as follows,

In introduction section

“Specifically, terpenoids, such as thymol and menthol, display bronchodilatory characteristics [1,2], relax airways, and ease respiratory discomfort. In addition, they regulate inflammatory pathways by suppressing the release of cytokines and the migration of leukocytes [3-5]. Moreover, phenols demonstrate antioxidant and anti-inflammatory properties by eliminating free radicals [6-8] and inhibiting pro-inflammatory enzymes such as cyclooxygenase [9,10]. Compounds like these reduce airway inflammation by reducing oxidative stress and cytokine production. These activities of phenols aid in easing respiratory issues by enhancing the functioning of lungs and mitigating the symptoms”

line 68 it is indicated that it inhibits oxidative stress, it is suggested to deepen this information.

Response: As suggested, we have included the essential information on oxidative stress in the revised manuscript. Updated text in the revised manuscript as follows,

In introduction section

“Interestingly, earlier studies suggest that Epigallocatechin gallate (EGCG), a significant catechin found in C. sinensis, boosts cellular antioxidant defenses by increasing the expression of antioxidant enzymes such as superoxide dismutase (SOD) and catalase [11]. These compounds synergistically act to protect cells and tissues from oxidative damage, contributing to the potential health benefits”

line 70 it is indicated that a richness in polyphenolic compounds so it is suggested to integrate more information of the compounds established for this mechanism.

Response: As suggested, we have included relevant information on polyphenols and its influence on AR. Updated text in the revised manuscript as follows,

In the introduction section,

“Moreover, previous reports suggest that polyphenols can impede the production of pro-inflammatory cytokines, such as interleukin-6 (IL-6), tumor necrosis factor-alpha (TNF-α), and interleukin-8 (IL-8). Consequently, it aids in reducing the inflammatory activities in the respiratory passages. Also, polyphenols can hinder the activation of nuclear factor-kappa B (NF-κB), a pivotal regulator of inflammation. This action contributes to lowering the magnitude of the inflammatory responses”

In the discussion section

Additionally, most of the positive implications of CSE can be attributed to the presence of procyanidins and other polyphenols like rutin and Baicalin/apigenin 7-glucuronide. The UPLC data suggest that CSE contains significant quantities of these bioactive compounds. In particular, compounds like rutin are recognized for their ability to defend against lung injury [12,13], while procyanidins aid in preventing inflammation in the airways [14,15]. Furthermore, procyanidin can enhance vascular health, which in turn helps to optimize the transportation of oxygen and reduce damage caused by inflammation [16]

Results section

This section provides a complete description of the studies carried out on the model and the effect of the different concentrations of the extract, covering different complementary techniques, and the results are clearly described. 

However, I consider a fundamental and important part to include is information on the phytochemical composition of C. sinensis from the background, as well as the characterization of the extracts obtained and tested in the animal model, in this way the results obtained can be discussed, although a discussion of the behavior of the different concentrations of the extracts is made, it is not clear which phytochemical compounds may be generating the effect.

Response: We thank reviewer for thoughtful comment on the CSE. As suggested, we have performed the chromatographic analysis of CSE to identify the functional components present in the extract. Updated information on these aspects in the revised manuscript as follows,

In material and methods

“Analysis CSE using UPLC-Q-TOF-MS

The CSE was analyzed to assess the presence of bioactive compounds using a UPLC-Q-TOF-MS system (Waters, Milford, MA, USA). The bioactive compounds in CSE were isolated with the Acquity UPLC BEH C18 column (Waters, Milford, MA, USA). The column was equilibrated with water containing 0.1% formic acid and eluted with a gradient of acetonitrile containing 0.1% formic acid. All the eluted samples were processed through the Q-TOF-MS system with positive electrospray ionization mode using the following instrument settings in the m/z 50 to 1500 range. In this assessment, the cone voltage was set at 20 V, while the capillary voltage was adjusted to 2.5 kV. The nebulized gas was set to 900 L/h at a temperature of 100 °C in positive mode, and the cone gas flow was set to 30 L/h. All analyses were performed using a locking spray to ensure accuracy and reproducibility. The compounds were identified based on online databases. The identified compounds were analyzed quantitatively employing the multiple reaction monitoring (MRM) mode of a UPLC-QTOF-MS system”

In result section,

Active componenets of CSE

A total of 6 compounds were identified. The bioactive compounds in CSE are Procyanidin B2, Procyanidin C1/C2, Rutin, Baicalin/apigenin 7-glucuronide, Diosmetin 7-glucuronide/6-O-methylscutellarin/Diosmetin 3-glucuronide, and a-Amylcinnamyl formate (Fig. 1B).

Figure 1B

references

  1. 1. Escobar, A.; Pérez, M.; Romanelli, G.; Blustein, G. Thymol bioactivity: A review focusing on practical applications. Arabian Journal of Chemistry 2020, 13, 9243-9269, doi:https://doi.org/10.1016/j.arabjc.2020.11.009.
  2. 2. Kim, T.; Song, B.; Cho, K.S.; Lee, I.S. Therapeutic Potential of Volatile Terpenes and Terpenoids from Forests for Inflammatory Diseases. Int J Mol Sci 2020, 21, doi:10.3390/ijms21062187.
  3. 3. de las Heras, B.; Hortelano, S. Molecular basis of the anti-inflammatory effects of terpenoids. Inflamm Allergy Drug Targets 2009, 8, 28-39, doi:10.2174/187152809787582534.
  4. 4. Del Prado-Audelo, M.L.; Cortes, H.; Caballero-Floran, I.H.; Gonzalez-Torres, M.; Escutia-Guadarrama, L.; Bernal-Chavez, S.A.; Giraldo-Gomez, D.M.; Magana, J.J.; Leyva-Gomez, G. Therapeutic Applications of Terpenes on Inflammatory Diseases. Front Pharmacol 2021, 12, 704197, doi:10.3389/fphar.2021.704197.
  5. 5. Ge, J.; Liu, Z.; Zhong, Z.; Wang, L.; Zhuo, X.; Li, J.; Jiang, X.; Ye, X.-Y.; Xie, T.; Bai, R. Natural terpenoids with anti-inflammatory activities: Potential leads for anti-inflammatory drug discovery. Bioorganic Chemistry 2022, 124, 105817, doi:https://doi.org/10.1016/j.bioorg.2022.105817.
  6. 6. Bocsan, I.C.; Magureanu, D.C.; Pop, R.M.; Levai, A.M.; Macovei, S.O.; Patrasca, I.M.; Chedea, V.S.; Buzoianu, A.D. Antioxidant and Anti-Inflammatory Actions of Polyphenols from Red and White Grape Pomace in Ischemic Heart Diseases. Biomedicines 2022, 10, doi:10.3390/biomedicines10102337.
  7. 7. Kruk, J.; Aboul-Enein, B.H.; Duchnik, E.; Marchlewicz, M. Antioxidative properties of phenolic compounds and their effect on oxidative stress induced by severe physical exercise. The Journal of Physiological Sciences 2022, 72, 19, doi:10.1186/s12576-022-00845-1.
  8. 8. Ruiz-Ruiz, J.C.; Matus-Basto, A.J.; Acereto-Escoffié, P.; Segura-Campos, M.R. Antioxidant and anti-inflammatory activities of phenolic compounds isolated from Melipona beecheii honey. Food and Agricultural Immunology 2017, 28, 1424-1437, doi:10.1080/09540105.2017.1347148.
  9. 9. Kelm, M.A.; Nair, M.G.; Strasburg, G.M.; DeWitt, D.L. Antioxidant and cyclooxygenase inhibitory phenolic compounds from Ocimum sanctum Linn. Phytomedicine 2000, 7, 7-13, doi:https://doi.org/10.1016/S0944-7113(00)80015-X.
  10. 10. Rahman, M.M.; Rahaman, M.S.; Islam, M.R.; Rahman, F.; Mithi, F.M.; Alqahtani, T.; Almikhlafi, M.A.; Alghamdi, S.Q.; Alruwaili, A.S.; Hossain, M.S.; et al. Role of Phenolic Compounds in Human Disease: Current Knowledge and Future Prospects. 2022, 27, 233.
  11. 11. Nagle, D.G.; Ferreira, D.; Zhou, Y.D. Epigallocatechin-3-gallate (EGCG): chemical and biomedical perspectives. Phytochemistry 2006, 67, 1849-1855, doi:10.1016/j.phytochem.2006.06.020.
  12. Tian, C.; Shao, Y.; Jin, Z.; Liang, Y.; Li, C.; Qu, C.; Sun, S.; Cui, C.; Liu, M. The protective effect of rutin against lipopolysaccharide induced acute lung injury in mice based on the pharmacokinetic and pharmacodynamic combination model. Journal of Pharmaceutical and Biomedical Analysis 2022, 209, 114480, doi:https://doi.org/10.1016/j.jpba.2021.114480.
  13. Chen, W.-Y.; Huang, Y.-C.; Yang, M.-L.; Lee, C.-Y.; Chen, C.-J.; Yeh, C.-H.; Pan, P.-H.; Horng, C.-T.; Kuo, W.-H.; Kuan, Y.-H. Protective effect of rutin on LPS-induced acute lung injury via down-regulation of MIP-2 expression and MMP-9 activation through inhibition of Akt phosphorylation. International Immunopharmacology 2014, 22, 409-413, doi:https://doi.org/10.1016/j.intimp.2014.07.026.
  14. Coleman, S.; Hurst, R.; Sawyer, G.; Kruger, M. The in vitro evaluation of isolated procyanidins as modulators of cytokine-induced eotaxin production in human alveolar epithelial cells. Journal of Berry Research 2016, 6, 115-124, doi:10.3233/JBR-160121.
  15. Tie, S.; Zhang, L.; Li, B.; Xing, S.; Wang, H.; Chen, Y.; Cui, W.; Gu, S.; Tan, M. Effect of dual targeting procyanidins nanoparticles on metabolomics of lipopolysaccharide-stimulated inflammatory macrophages. Food Science and Human Wellness 2023, 12, 2252-2262, doi:https://doi.org/10.1016/j.fshw.2023.03.045.
  16. Chen, F.; Wang, H.; Zhao, J.; Yan, J.; Meng, H.; Zhan, H.; Chen, L.; Yuan, L. Grape seed proan thocyanidin inhibits monocrotaline-induced pulmonary arterial hypertension via

Round 2

Reviewer 3 Report

Comments and Suggestions for Authors

The authors adequately addressed the raised comments; thanks. 

Comments on the Quality of English Language

Minor editing of the English language is required. 

Author Response

REBUTTAL LETTER

We would like to express our sincere thanks to the reviewers who identified areas of our manuscript that needed corrections/formations/ modifications. We have improved our manuscript substantially as per the reviewer comments.

Reviewer 3

Minor editing of the English language is required. 

Response: Thank you for the observations. The revised manuscript is proofread by a fluent speaker of English. We believe that the newly revised manuscript is free of linguistic errors.

Reviewer 4 Report

Comments and Suggestions for Authors

Dear authors, I thank you for having taken into account the suggestions made to highlight the importance of the research carried out, I consider the new study of the chemical composition of the extracts to be very appropriate, however I consider that it should not be incorporated in the same image as the animal model, the image of the animal model should be included in the section where the results of this model begin to be described and the description should talk a little about it.

That is to say, I suggest that the first image should be the part of the description of the compounds of the extracts and in the second image the biological model should be included in some section, in this way the information is clearer, because in this way it is disarticulated. 

Author Response

REBUTTAL LETTER

We would like to express our sincere thanks to the reviewers who identified areas of our manuscript that needed corrections/formations/ modifications. We have improved our manuscript substantially as per the reviewer comments.

Reviewer 4

However I consider that it should not be incorporated in the same image as the animal model, the image of the animal model should be included in the section where the results of this model begin to be described and the description should talk a little about it.

That is to say, I suggest that the first image should be the part of the description of the compounds of the extracts and in the second image the biological model should be included in some section, in this way the information is clearer, because in this way it is disarticulated. 

Response: Thank you for your valuable suggestion. As suggested, we have updated the manuscript so that the first image (Figure 1) describes the compound analysis, and the second image (Figure 2) includes the experimental design (Biological model).